# Real-Life Use Patterns of Androgen Receptor Pathway Inhibitors (ARPIs): A Nationwide Register-Based Study in Finland During 2012–2023

**DOI:** 10.3390/cancers17193162

**Published:** 2025-09-29

**Authors:** Terhi Kurko, Pekka Heino, Pirkko-Liisa Kellokumpu-Lehtinen, Kati Sarnola, Hanna Koskinen, Maarit Bärlund

**Affiliations:** 1Research Unit, The Social Insurance Institution of Finland, 00056 Helsinki, Finland; pekka.heino@kela.fi (P.H.); kati.sarnola@kela.fi (K.S.); hanna.koskinen@kela.fi (H.K.); 2Faculty of Medicine and Health Technology, Tampere University, 33100 Tampere, Finland; pirkko-liisa.kellokumpu-lehtinen@tuni.fi (P.-L.K.-L.); maarit.barlund@pirha.fi (M.B.); 3Research, Development and Innovation Centre, Tampere University Hospital, The Wellbeing Services County of Pirkanmaa, 33101 Tampere, Finland; 4Department of Oncology, Tays Cancer Centre, Tampere University Hospital, The Wellbeing Services County of Pirkanmaa, 33101 Tampere, Finland

**Keywords:** androgen pathway inhibitors (ARPIs), prostate cancer, register study, sequential use, use patterns

## Abstract

Androgen pathway inhibitors (ARPIs) are widely used in prostate cancer. We conducted a register data-based analysis assessing their use and especially sequential use, e.g., use of one medicine after another, in Finland by using a dataset covering the whole country during 2012–2023. We found that one-third of patients use at least two ARPIs sequentially to inhibit testosterone effect. The total cost of sequential use was nearly €44 million. However, there are no large clinical trials published demonstrating the benefits of sequential treatment. More evidence is needed to justify sequential use.

## 1. Introduction

Prostate cancer (PC) is the most prevalent cancer among males in the Western World. In 2022, nearly 1.5 million men worldwide were diagnosed with PC, including 5514 cases in Finland [1,2]. Key causes of PC, similar to many other cancers, include genetic factors, exposure to different carcinogens, and increasingly, lifestyle-related factors [3]. The rapid increase in cases is attributed to the ageing population, early detection through prostate-specific antigen (PSA) testing, and advancements in imaging modalities [4]. Despite the generally good survival rates for patients with PC, approximately 10–20% of all PC patients develop castration-resistant prostate cancer (CRPC), the most common lethal form of PC, and, like the most aggressive PC, neuroendocrine prostate cancer (NEPC) predicting a poor outcome [5,6,7]. Worldwide, 396,792 PC-related deaths were reported in 2022, with 920 of those occurring in Finland [1,2].

In metastatic hormone-sensitive prostate cancer (mHSPC), androgen deprivation treatment (ADT) has been the standard treatment for decades. Eventually, mHSPC will progress to castration-resistant prostate cancer (mCRPC) despite the ADT therapy [6]. Until 2010, docetaxel was the only life-prolonging agent for mCRPC [7,8,9]. However, with the introduction of androgen receptor pathway inhibitors (ARPIs), first abiraterone, then enzalutamide, and more recently apalutamide and darolutamide, the landscape of mCRPC treatment has changed dramatically. Abiterone belongs to the first generation of ARPIs and enzalutamide, darolutamide, and apalutamide to the second generation [10]. They all competitively and reversibly inhibit the binding of testosterone and 5α-dihydrotestosterone (DHT) to the ligand binding domain of androgen receptor, but the second-generation medicines also downstream inhibition of AR translocation to nucleus from cytoplasm, recruitment of coactivators, and binding to DNA. A detailed illustration on the mechanism and significance of ARPIs is presented in [10], for instance. Initially, abiraterone and enzalutamide were approved for use after docetaxel. Later, abiraterone and enzalutamide were approved for first-line use before docetaxel. They have since been progressively used in earlier stages, such as high-risk nonmetastatic castration-resistant prostate cancer (nmCRPC) and metastatic hormone-sensitive prostate cancer (mHSPC) [11].

In the second-line treatment of mCRPC, cabazitaxel improved overall survival, OS, (HR 0.70; 95% CI 0.59–0.83) compared with mitoxantrone in patients previously treated with docetaxel, and its biweekly dosing also seems to be well tolerated [12,13]. The CARD trial (NCT02485691) compared cabazitaxel with a second ARPI. The median OS was 13.6 months with cabazitaxel and 11.0 months with the second ARPI (HR 0.64; 95% CI 0.46–0.89; *p* = 0.008) [14]. A second-line ARPI (abiraterone for patients previously treated with enzalutamide and vice versa) had only modest beneficial activity in a phase 2 trial (NCT02125357, https://clinicaltrials.gov/show/NCT02125357, accessed on 15 June 2025) and in several real-world studies [15,16,17]. In addition, there is substantial evidence indicating cross-resistance between abiraterone and enzalutamide, and thus the use of a second ARPI (abiraterone after enzalutamide or vice versa) is not recommended in ESMO (European Society for Medical Oncology) Clinical Practice Guidelines [18,19]. However, EAU (European Association of Urology) guidelines do not recommend sequencing of ARPIs if the time of response to the first ADT and the first ARPI was short, and high-risk features of rapid progression are present [20]. The optimal sequence or combination of these agents remains largely unknown.

Annual healthcare costs per patient for metastatic prostate cancer are substantial and have increased over time, corresponding to the approval of new oral therapies used in treating metastatic prostate cancer [4,5,6]. Today, the number of PC patients is constantly increasing, and the patients live longer, leading to rapidly rising costs of cancer treatments and, consequently, challenges in access to treatment worldwide [21,22]. Therefore, it is important to assess if there exists low-value cancer care practises. Given the high prices of ARPIs, evidence on the real-world utilisation and costs of ARPIs is limited, especially regarding the sequencing of ARPIs in Scandinavia [23]. To our knowledge, we lack knowledge on the population level on ARPI use patterns in real life, especially the use patterns which may have only little clinical benefit but high costs. To be better prepared to face these facts, we conducted a register-based study. The aim of this nationwide study was to assess ARPI use patterns, especially sequential use and treatment costs, in 2012–2023.

## 2. Patients and Methods

All permanent residents in Finland are covered under the Finnish National Health Insurance (NHI) scheme, implemented by the Social Insurance Institution of Finland (SII) [9]. Patients are entitled to reimbursements for outpatient medicines confirmed as reimbursable under the NHI scheme by the Finnish Pharmaceutical Pricing Board (PPB). Since 2012, ARPIs have been included in the Finnish reimbursement scheme as follows: abiraterone in 2012, enzalutamide in 2014, and apalutamide and darolutamide in 2020. The detailed information concerning their uptake in the Finnish reimbursement system is presented in Appendix A.

### 2.1. Description of the Used Register Data and Definitions of the Key Studied Patterns

We used administrative register data on all ARPI purchases reimbursed under the NHI scheme and recorded in the dispensations reimbursable under the NHI scheme register maintained by SII. From the register, we extracted information on all ARPI purchases (enzalutamide; apalutamide; darolutamide; and abiraterone) [22] made between January 2012 and December 2023. We collected information on the purchase date, volume of the purchase in defined daily doses (DDD), and number of purchases per year per patient as well as a unique pseudonymised patient identifier and age of the patient purchasing ARPIs [24]. In addition, we collected data on the starting date of the right to reimbursement at a special rate for prostate cancer for all men purchasing ARPIs, costs of ARPIs, and possible time of death from the NHI registers.

For this study, a new ARPI user was defined as a person who had not purchased any observed ARPIs for 365 days prior to the purchase. The first purchase was identified for each of the medicines, and the follow-up of the users started from the first purchase. The duration of individual ARPI treatment episodes was calculated as the difference between the first and last purchase of the studied medicine. For sequential use, it was calculated as the difference between the first purchase date of the sequentially used medicine and the last purchase date of the previously used medicine. Sequential use was defined as the purchase of different ARPIs one after another. If there were no purchases within 365 days, the use was considered discontinued. Additionally, we merged data concerning the possible death of the initiators. We conducted Kaplan–Meier analysis to compare the median overall survival (OS) among patients with single ARPI use and sequential ARPI use.

Based on the median duration of sequential use for each medicine, the number of users and used medicine packages, and the retail prices of each medicine package, we calculated the average costs for sequential use during the study period (Appendix A).

### 2.2. Ethics Statement

As the study was based on secondary register data, no ethics board approval was required (Finnish National Board on Research Integrity TENK) according to Finnish legislation. SII approved the use of the data for the current study. The data used in the study were fully pseudonymised before the authors accessed the data. All data preparation and linkage in the study were performed with pseudo-identifiers, and the authors did not have access to information that could identify individual participants at any stage of the study. Legal restrictions prevent the open sharing of the pseudonymised data supporting the current study, as individual-level health data are considered highly sensitive, and access is strictly regulated by law in Finland (Act on Secondary Use of Health and Social Data (552/2019)). Interested parties may, however, apply for permission to access the data from Findata (https://findata.fi/en/, accessed on 15 June 2025).

## 3. Results

Altogether, 8369 patients initiated the use of reimbursed ARPIs in Finland during 2012–2023 (Table 1). The number of users steadily increased over time, reaching its peak in the year 2023, when a total of 1313 patients initiated the treatment (Figure 1). The most used medicine was enzalutamide with 3149 users. The median age of users at the time of the initiation was 75.1 years (range 41.0–98.1 years). Throughout the study period, the median age of patients at the time of treatment discontinuation increased modestly, rising from 71.0 years in 2012 to 76.8 years in 2023. During the observed period, 61% of the patients (*n* = 5091) had died by the end of 2023.

Throughout the study period, the median duration of first-line medicine use was longest for darolutamide (12.1 months) and shortest for abiraterone (6.6 months) (Table 1). However, the median duration of use decreased for all medicines when used sequentially.

### 3.1. Patterns Related to Sequential ARPI Use

Nearly a third of the patients (*n* = 2685, 32.1%) used at least two ARPIs sequentially (Table 1, Figure 2). The two most common patterns of sequential use were treatment initiation with abiraterone, followed by enzalutamide (14.9% of the patients), and initiation with enzalutamide, followed by abiraterone (14.5%). The median time between the first purchase of the second medicine and last purchase of the first medicine was approximately 1.5 months, i.e., 42 days. The number of users initiating sequential use within 42 days was 1354 men (16.2% of all men initiating ARPIs during our study period), and the number of users initiating sequential use in 43 days or later was 1321 men (15.8%). Sequential use, in general, and in 42 days or less was most common in years 2018–2019, especially in the year 2018 (Figure 3). The most sequentially used medicine was abiraterone (with 1354 patients), and the median duration of sequential use was longest for darolutamide (342 days) and shortest for apalutamide (106 days). Nearly all the sequential use combinations were combining abiraterone and enzalutamide (96%). There was virtually no difference in the median age between users with and without sequential use (75.9 vs. 76.7 years during the whole study period).

During the study period, the proportion of initiators whose initiation led to sequential use first increased and then decreased (Figure 2). In 2012, over a third (36%) of all treatment initiations led to sequential use. This proportion rose to over half (56%) in 2017 but then decreased to 14% in 2022.

During our study period, 51% of the patients using single ARPI and 82% of the patients with sequential ARPI use died. Based on Kaplan–Meier method, the median OS among single ARPI users was 11.8 months and among sequential users 25.7 months.

### 3.2. Costs of Hormonal Prostate Cancer Medicines and Sequential ARPI Use

Between 2012 and 2023, ARPI costs amounted to €453.3 million in Finland. This includes €278.2 million for enzalutamide, €137.8 million for abiraterone, €26.7 million for apalutamide, and €10.6 million for darolutamide. During the study period, the number of new ARPI users increased from 139 in 2012 to 1452 persons in 2023 (Figure 1). During the study, the cost of hormonal medicines for prostate cancer surged by over 400% (Figure 4). In 2023, the cost of ARPIs reached €66 million. Meanwhile, the cost of older hormonal medicines saw a moderate increase in just 12% over the study period. Overall, the costs of sequential use amounted to €43.8 million, representing 9.7% of the total costs and medicine use of 2685 patients (Appendix A).

**Figure 4 cancers-17-03162-f004:**
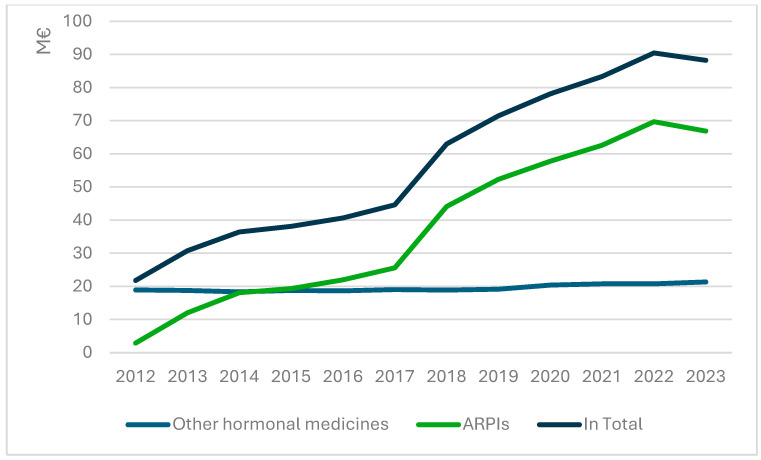
Trends in the costs of hormonal medicines for prostate cancer treatment over the years. ARPIs are shown with a green line, other hormonal medicines for prostate cancer with a light blue line, and total cost with a dark blue line.

## 4. Discussion

According to this national analysis, ARPIs were widely adopted for treating prostate cancer patients in Finland after their inclusion in the reimbursement system. Nearly 8400 patients initiated ARPI use during our study period, and approximately one-third of them (32%) used at least two ARPIs sequentially. Median age of the users was 75 years. During our observation period, the proportion of treatment initiations leading to sequential use increased first and later decreased, being 14% in 2022. Costs of sequential use were nearly €44 million.

Initially, reimbursement of ARPIs was limited to treating mCRPC patients whose cancer progressed during or after docetaxel-based chemotherapy. Later, it was extended to asymptomatic or mildly symptomatic mCRPC patients after androgen deprivation therapy failure, increasing ARPI usage and costs. During the observed years, hormonal medicine costs for prostate cancer rose exponentially due to ARPIs, from €2.9 million in 2012 to €66.0 million in 2023. The increased use of ARPIs is also related to the increasing number (increase of 18%) of new prostate cancer patients (from 4668 patients in 2012 to 5,514 patients in 2022) [2]. Similarly, the proportion of prostate cancer survivors rose by 40% (from 43,579 in 2012 to 61,154 in 2022). This is due to the ageing population, earlier diagnosis, and better treatments such as ARPIs. On the other hand, this development also burdens society and medical experts. The Finnish healthcare system, founded on mainly publicly funded healthcare services, necessitates a critical assessment of treatment effectiveness.

Surprisingly, sequential use of ARPIs was instantly adopted into clinical practice without any evidence-based research. The publication of a phase II study in 2019 (NCT02125357, https://clinicaltrials.gov/show/NCT02125357, accessed on 15 June 2025) and several real-world studies accelerated sequential use, peaking in 2019–2020 [15]. Despite no benefit shown in randomised phase III trials, one-third of the patients in our study received second-line or subsequent ARPIs. This raises a critical question: why were these agents used sequentially despite the lack of proven efficacy? Possible explanations include avoiding of end-of-life discussions, e.g., referral to palliative care after progression to one ARPI. Additional explanations could be influence from pharmaceutical marketing, management of adverse events, or perhaps a lack of awareness regarding current treatment guidelines. It is well known that today oncologist and urologists are inundated with information of new cancer medicine, clinical trial data, and repeatedly updated treatment guidelines, which all together make it difficult to stay up to date, especially when national healthcare system covers the costs.

Over 11 years, reimbursement costs for these treatments amounted to €43.8 million, despite the short duration of sequential medicine use (about 6 months). This practice contradicts the ESMO guideline, which does not recommend sequential use of ARPIs [18,19]. Additionally, the EAU guidelines do not directly endorse ARPIs [20]. We found that nearly all (over 96%) of the combinations used sequentially were combinations of abiraterone and enzalutamide, while combinations of second-generation ARPIs were very rare. We also found that starting in 2020, there was a significant change in the ARPIs used, and also sequential use diminished, which may be explained by the uptake of the newest ARPIs, darolutamide and apalatumide, and also the updated ESMO guidelines [18,19].

This analysis raised concerns about age, as the oldest patient was 98 years old, and the median age was 75.1 years. High prevalence of comorbidities and polypharmacy among mCRPC patients on ARPIs may lead to medicine–medicine and medicine–comorbidity interactions, affecting outcomes [25,26,27,28]. ARPIs have potential risks of adverse effects (AEs) such as fatigue, cognitive impairment, and functional decline, leading to falls and a poor quality of life [10,16]. Androgen withdrawal with ARPIs affects cardiovascular health, as shown in a recent meta-analysis [28]. Worsening of cognitive function with ADT has been shown over decades, raising concerns about costs and patient outcomes. We also found shorter ARPI use per patient than in randomised first-line trials. These findings emphasise the need for meticulous patient selection, counselling, optimisation, and monitoring during therapy. In addition, AEs may be higher in real-world settings. Therefore, best supportive care may serve the patient’s overall wellbeing better than the initiation of a novel treatment with AEs.

Data on ARPIs’ efficacy in older adults are limited to those with excellent performance status and almost normal organ functions. The Advanced Prostate Cancer Society (APCCC) 2022 panellists agreed that geriatric assessment (GA) should be performed on patients older than 75 years, and International Society of Geriatric Oncology (SIOG) guidelines recommend GA for patients over 75 years with many comorbidities [20,29]. Frailty and specific comorbidities significantly influence cancer control outcomes in mCRPC patients [27]. Evaluating ARPIs’ AE profile in the context of GA is necessary to ensure safety in older adults with prostate cancer [27,28]. Real-world data showed shorter ARPI use in Finnish PC patients compared to the prospective clinical trials, as shown in a Swiss study [30]. High costs of PC treatments necessitate considering the patients’ life expectancy [31]. Preventing futile treatments in palliative patients enables better symptom management without the ARPI side effects, such as anaemia, hot flushes, hypertension, fatigue, musculoskeletal pain, for instance, [10,32] reducing the financial impact of cancer care. In addition to adverse events, drug interactions, drug resistance, and impact on functional capacity should also be taken into account as major limitations of ARPI treatment.

Our findings can inform reimbursement policies and clinical decision-making by offering an example why paying attention to rational use of medicines and especially to pharmacotherapy with low value care but high costs is important. Based on the statistics of SII, ARPIs have been for many years among medicines with the highest reimbursement costs in the whole Finland due to their high prices and relatively large number of users. Therefore, it is especially important to pay attention that high-cost medicines are used in a way that benefits frail patients most.

Limitations of our retrospective study include not knowing at which phase of the disease ARPIs were used, what the previous prostate cancer treatments such as surgical resection/radioactive seed implantation and comorbidities were, and what performance status of the patients was. Adverse events were unavailable, but a short duration of use may indicate response and adverse events. The utilised register does not contain any information on overlapping prescriptions or treatment switches within short periods, for instance due to adverse event, but based our assessment treatment, switches within a short period were rare. We also missed register data concerning the cause of mortality in this study. Therefore, a future study is needed to assess in detail how survival outcomes and life expectancy are related to different ARPI use patterns. In addition, it would be worthwhile to assess how do patients with or without sequential ARPI use differ. The strengths include the inclusion of all ARPI users and their purchases nationwide recorded in the studied register.

## 5. Conclusions

Our study, based on nationwide longitudinal register data, found that sequential ARPI use is common and costly. This treatment schedule in Finnish PC patients does not follow the recent guidelines by several European organisations. Our findings on ARPI use patterns and patient selection highlight the need to more critically assess patients who may benefit from ARPI treatment the most.

## Figures and Tables

**Figure 1 cancers-17-03162-f001:**
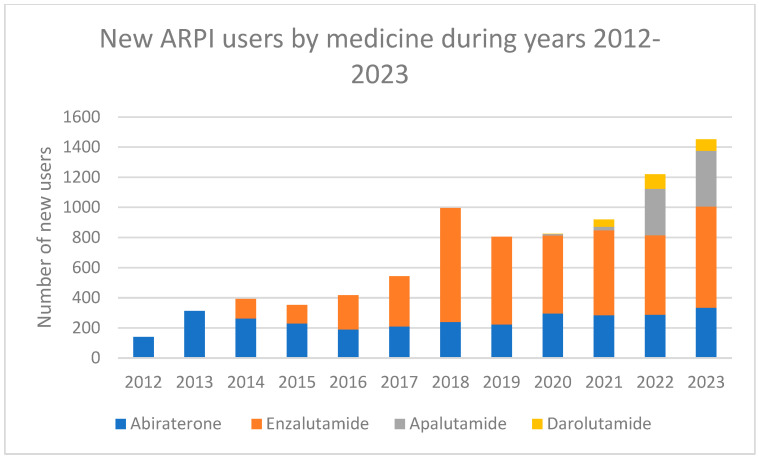
New Androgen Receptor Pathway Inhibitor users  during 2012–2023 by medicine.

**Figure 2 cancers-17-03162-f002:**
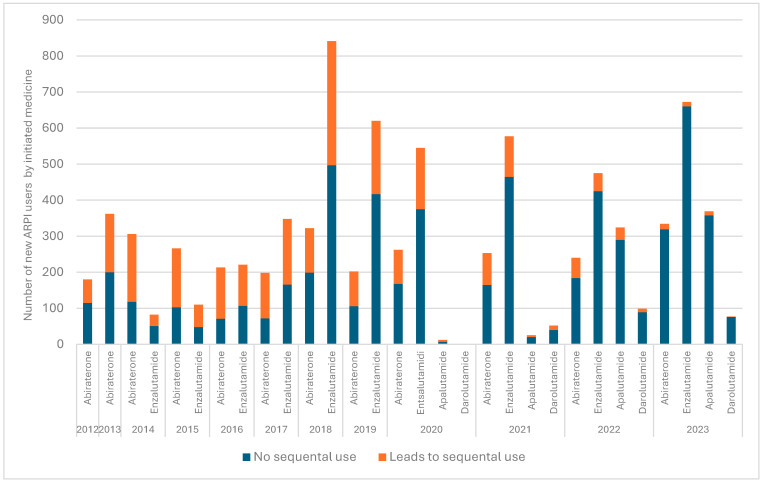
Number of individual users of different ARPIs in Finland from 2012 to 2023, categorised by medicine and sequential use (orange line) or non-sequential use (blue line).

**Figure 3 cancers-17-03162-f003:**
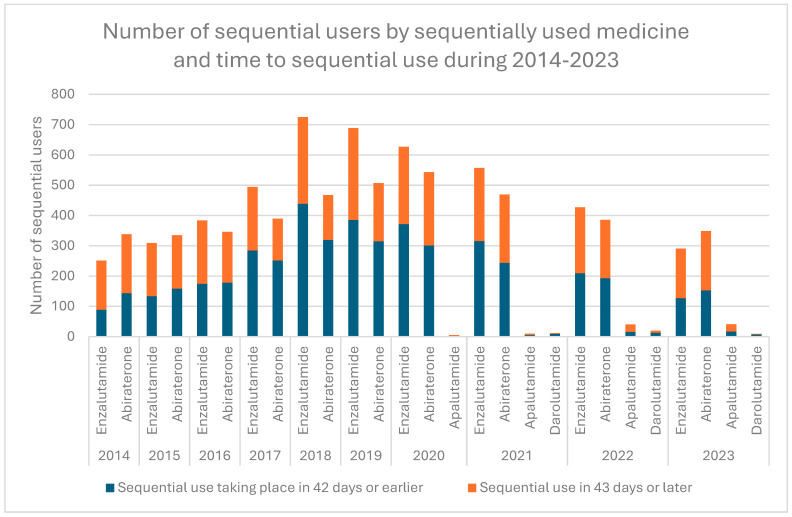
Annual count of sequential users by medicine divided by median time (42 days) to sequential use.

**Table 1 cancers-17-03162-t001:** Description of ARPI users and the duration of treatment use patterns.

**Number of Patients Initiating Treatment**	**8369**			
Median age of the users, years at the time of treatment initiation (range)	75.1 (41.0–98.1)			
**Number of ARPI users who used**	*n*		%	
One ARPI	5684		67.9%	
Two ARPIs	2537		30.3%	
Three ARPIs	134		1.6%	
Four ARPIs	14		0.2%	
**Patterns related to the first line of treatment**				
The medicine chosen in the first line	*n*		*%*	
Abiraterone	2977		35.6	
Enzalutamide	4438		53.0	
Apalutamide	727		8.7	
Darolutamide	227		2.7	
**Treatment duration characteristics**				
Median duration of first-line use	Months			
Overall	8.7			
Leads to sequential use	8.3			
Not leading to sequential use	9.6			
Median duration of first-line use by medicine	Months			
Abiraterone	6.6			
Enzalutamide	10.2			
Apalutamide	9.3			
Darolutamide	12.1			
**Use patterns related to sequential use**				
Time to sequential use *	Months			
Median	1.4			
Minimum	0			
Maximum	45.1			
Median duration of sequential use by medicine used sequentially (users)	Months			
Abiraterone (n = 1354)	6.4			
Enzalutamide (n = 1321)	6.9			
Apalutamide (n = 5)	3.5			
Darolutamide (n = 5)	11.4			
**Observed combinations in sequential use**	users, n	% of all users		% of all sequential users
Abiraterone- enzalutamide	1249	14.9		47%
Enzalutamide-abiraterone	1214	14.5		45%
Enzalutamide- abiraterone-enzalutamide	63	0.8		2%
Abiraterone- enzalutamide-abiraterone	61	0.7		2%
Other combinations (altogether 14 different ones) **	98	1.2		4%

* Calculated from the start date of the second medicine until the stop date of the first medicine. ARPIs = androgen pathway inhibitors. ** Combination of second-generation ARPIs took place among 15 patients (0.6%).

## Data Availability

Legal restrictions prevent the open sharing of the pseudonymised data supporting the current study, as individual-level health data are considered highly sensitive, and access is strictly regulated by law in Finland (Act on secondary use of health and social data (552/2019)). Interested parties may, however, apply for permission to access the data from Findata (www.findata.fi, accessed 15 June 2025).

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
