# Peer review of "Real-Life Use Patterns of Androgen Receptor Pathway Inhibitors (ARPIs): A Nationwide Register-Based Study in Finland During 2012–2023"

_cancers, 2025, doi:10.3390/cancers17193162_

Round 1

Reviewer 1 Report

Comments and Suggestions for Authors

The current manuscript focuses on the cost of androgen pathway inhibitors for the patients with pancreatic cancers. The study reads well, and the study design is fluent, yet additional analysis would be beneficial to capture the interest of the audience. 

  1. The increasing cost for the drugs is associated with the increasing unit price or with the patients. Therefore, the unit price and number of patients receiving the drugs need to be included.
  2. In Figure 1, x labels are overlapping. Please correct it.

Author Response

Referee 1 Comments and Suggestions for Authors

The current manuscript focuses on the cost of androgen pathway inhibitors for the patients with pancreatic cancers. The study reads well, and the study design is fluent, yet additional analysis would be beneficial to capture the interest of the audience. 

Response: Thank you for your kind feedback!

  1. The increasing cost for the drugs is associated with the increasing unit price or with the patients. Therefore, the unit price and number of patients receiving the drugs need to be included.

Response: Thank you for pointing this important aspect out! Based on your excellent comment, we have introduced a new figure 1, to present number of new users by each ARPIs by medicine. In addition we have introduced the following sentence to the manuscript, please see lines 218–220:

“During the study period, the number of new ARPI users increased from 139 in 2012 to 1,452 persons in 2023 (Supplementary Figure 1).”

In addition, while introducing the new Figure 1, we have renumbered all the Figures to 2–5 and also in the main text where we abbreviate to them.

2. In Figure 1, x labels are overlapping. Please correct it.

Response: Thank you for this good comment! We have now corrected the labels of x-axis in current

                    Figure 2 (previous Figure 1).

Reviewer 2 Report

Comments and Suggestions for Authors

In the submitted manuscript authors presented a pattern of androgen receptor pathway inhibitors (ARPIs) usage among prostate cancer (PC) patients in Finland during the period 2012 – 2023, and showed that one-third of patients used at least two ARPIs sequentially, what costed nearly €44 million, while there are actually no large clinical trials published demonstrating the benefits of sequential treatment.

This manuscript is interesting and presented data potentially useful. It is quite well written; however, there are several things which must be corrected and further improved.

1) Lines 51-52: The most aggressive form of PC is actually a neuroendocrine prostate cancer (NEPC), which can arise de novo (rarely) or more often as treatment-emergent NEPC (t-NEPC) after long-term androgen-deprivation therapy or AR pathway inhibitors (enzalutamide, abiraterone).

2) All abbreviations, like ESMO, EAU, APCCC and SIOG, must be explained. Also, all abbreviations mentioned in tables and supplementary tables must be explained in that table's footnote. In addition, there is no need to explain the same abbreviation multiple times like you did in Supp. Table 1.

3) Line 104: It is unclear what sex of a patient has with a purchasing of drugs for prostate cancer?!

4) Line 125: Since you gain an approval, state the identification number of the official document.

5) Line 133: Explain what is Findata and provide its web address (where anyone could apply for a permission).

6) Line 141: Whenever you state a percentage in the text, also provide its absolute number (count) in parentheses.

7) Table 1 needs several improvements:

a) Stating a median value in both months and days is meaningless. If you have collected data in months, then instead of median days, provide a range in months. This information would be more useful.

b) It is unclear why only "Time to sequential use" was presented with both median and mean values?!

c) For "The most common combinations in sequential use" provide ALL observed combinations, since in present form provided data is not completely informative. Also, "users, n=" correct to just "n".

8) Text in lines 200-203 seems more like a result, not discussion. However, it would be very informative if you could create an overlay chart in which you would graph incidence, mortality and annual count of sequential ARPIs users during each year of the period 2012 – 2023.

9) Whenever a trial has been mentioned, its reference or ClinicalTrials.gov ID must be provided (e.g., in lines 211 and 225).

10) 'Introduction' lacks more information about what all types of drugs ARPIs are, since abiraterone and enzalutamide are not of the same type. Also, more both information and discussion should be provided if the sequence of different types of ARPIs are important for PC treatment, and how that relates to your results.

11) The reference list is a complete mess, since the style of almost each reference differs, and is not what MDPI requires! Also, for an internet resource, it is important to provide a date when it was last accessed, NOT cited!

Author Response

Referee 2 Comments and Suggestions for Authors

In the submitted manuscript authors presented a pattern of androgen receptor pathway inhibitors (ARPIs) usage among prostate cancer (PC) patients in Finland during the period 2012 – 2023, and showed that one-third of patients used at least two ARPIs sequentially, what costed nearly €44 million, while there are actually no large clinical trials published demonstrating the benefits of sequential treatment.

This manuscript is interesting and presented data potentially useful. It is quite well written; however, there are several things which must be corrected and further improved.

1) Lines 51-52: The most aggressive form of PC is actually a neuroendocrine prostate cancer (NEPC), which can arise de novo(rarely) or more often as treatment-emergent NEPC (t-NEPC) after long-term androgen-deprivation therapy or AR pathway inhibitors (enzalutamide, abiraterone).

Response:  Thank you! We totally agree that NEPC is the most aggressive entity of PC right from the beginning and could arise “de no” after long ADT treatment, which is assumed to happen in 17% to % of CRPC cases. In that kind of situation platin based chemotherapy is the treatment of choice. However, still the “normal” CRPC is more common and may still react to ARPIs, but will lead to metastatic disease and death. Thus, we have changed the sentence in previous line 51-52, current lines 52–55 according to this notion as follows

 “approximately 10%–20% of all PC patients develop castration-resistant prostate cancer (CRPC), the most common lethal form of PC, and thus like the most aggressive PC, neuroendocrine prostate cancer (NEPC) predicting a poor outcome [5–7].”

2) All abbreviations, like ESMO, EAU, APCCC and SIOG, must be explained. Also, all abbreviations mentioned in tables and supplementary tables must be explained in that table's footnote. In addition, there is no need to explain the same abbreviation multiple times like you did in Supp. Table 1.

Response: Thank you for this good comment. We have explained now all the abbreviations and also modified the supplementary table 1 accordingly.

3) Line 104: It is unclear what sex of a patient has with a purchasing of drugs for prostate cancer?!

Response: Thank you! The sex of patient was included to guarantee that all dispenses recorded in registers are correct. Based on your good comment we have deleted term sex in current line 123 (previous 104).

4) Line 125: Since you gain an approval, state the identification number of the official document.

Response: Thank you so much for this good comment! The Finnish Social Insurance Institution (KELA) gave the approval to conduct this register-based study. The identification number for this decision is KELA/ DARK/ 2/500/2022.

5) Line 133: Explain what is Findata and provide its web address (where anyone could apply for a permission).

Response: Thank you for this good comment. Findata is the abbreviation for Finnish Social and Health Data Permit Authority. Findata is the centralized data permit authority, from which the use of register-based individual level data for secondary use can be applied. Based on your excellent comment we have added the internet-site of Findata in the line 153 in the manuscript as follows

 Interested parties may, however, apply for permission to access the data from Findata (https://findata.fi/en/).

6) Line 141: Whenever you state a percentage in the text, also provide its absolute number (count) in parentheses.

Response: Thank you! The absolute number has been added in parentheses (current line 162).

7) Table 1 needs several improvements:

  1. a) Stating a median value in both months and days is meaningless. If you have collected data in months, then instead of median days, provide a range in months. This information would be more useful.

Response: Thank you, We have deleted now median values presented in days

  1. b) It is unclear why only "Time to sequential use" was presented with both median and mean values?!

Response: Thank you for this good comment! This was done just to better describe the skewness of the data but based on your comment we have deleted means, as median describes these patterns in a better way.

  1. c) For "The most common combinations in sequential use" provide ALL observed combinations, since in present form provided data is not completely informative. Also, "users, n=" correct to just "n".

Response: Thank you for this good comment! We have now introduced a new row in the table 1 presenting also the other found combinations

8) Text in lines 200-203 seems more like a result, not discussion. However, it would be very informative if you could create an overlay chart in which you would graph incidence, mortality and annual count of sequential ARPIs users during each year of the period 2012 – 2023.

Response: Thank you so much for this good suggestion! Based on your comment, we conducted a very crude Kaplan Maier analysis assessing the overall survival comparing patient using ARPIs sequentially and using them in monotherapy. We added the following paragraphs in the methods and results section, as follows:

Methods (lines 136–137):  We conducted Kaplan-Meier analysis to compare the median overall survival (OS) among patients with single ARPI use and sequential ARPI use.

Results lines (190–192): During our study period of the patients using single ARPI 51% and of the patients with sequential ARPI use 82% had died. Based on Kaplan-Maier method the median OS among single ARPI users was 11.8 months and among sequential users 25.7 months.

A limitation of our study is that we did not have in use register-based data containing information on the cause of mortality. As the majority of the patients included are older men with probably many comorbidities, we did not assess more detailed survival patterns, as the possible cause of death could be many other in addition to prostate cancer. Based on your valuable comment we have included the following sentence in the discussion section of the manuscript, please see lines 315–317:

“.  We also missed the register-data concerning the cause of mortality in this study. Therefore, a future study is needed to assess how more detailed survival outcomes and life expectancy are related to different ARPI use patterns. “

9) Whenever a trial has been mentioned, its reference or ClinicalTrials.gov ID must be provided (e.g., in lines 211 and 225).

Response: Thank you for this good notification! These have been added in to the manuscript.

10) 'Introduction' lacks more information about what all types of drugs ARPIs are, since abiraterone and enzalutamide are not of the same type. Also, more both information and discussion should be provided if the sequence of different types of ARPIs are important for PC treatment, and how that relates to your results.

Response: Thank you! Based on your comment we have added the following sentences in the introduction

Abiterone belongs to first generation of nonsteroidal androgen receptor antagonist and enzalutamide, darolutamide and apalutamide to second generation. They all competitively and reversibly inhibit binding of testosterone and DHT to ligand binding domain of androgen receptor, but the second-generation medicines also downstream inhibition of AR translocation to nucleus from cytoplasm, recruitment of coactivators and binding to DNA [10]. (please see lines 65-71)

In addition, into the discussion we have added the following sentence, please see lines 266–268.

 We found that nearly all (over 96 %) of all the combinations used sequentially were combinations of abiraterone and enzalutamide and combinations of second-generation ARPI were very rare.

11) The reference list is a complete mess, since the style of almost each reference differs, and is not what MDPI requires! Also, for an internet resource, it is important to provide a date when it was last accessed, NOT cited!

Response: We apologize the mess! We have now updated the reference list to take into account new references introduced based on the comments obtained and also modified the reference list to follow the instructions of MDPI.

Reviewer 3 Report

Comments and Suggestions for Authors

This manuscript reports a study on the use of ARPI in the treatment in CRPC in cancer patients. The study is potentially interesting but the enthusiasm for the topic is significantly limited by the manner the data were reported and discussed. Without a proper clarification and discussion of the points raised below no clear assessment of the significance of the results reported here can be properly made.

Comments:

  1. The authors reports that in their country the sequential use of ARPI is not recommended but yet, it appears that ARPI were used sequentially in all the years covered by the study. This begs the question: if the sequential use is not recommended, why these therapeutic agents were used sequentially? Insurgence of resistance? Recurrence of the cancer? Other reasons?
  2. Starting in 2020, there is a significant change in the ARPIs used (still sequentially, apparently) for the treatment of CRPC. Some explanation for the observed switch should be provided,
  3. Where all the patients affected by CRPC placed on sequential administration of ARPI? If not, which patients were placed in this treatment category? And why? Did any of the patients considered here remain on a single ARPI treatment? And if so, what was the reason or rational for doing so?
  4. Did any percentage of the patients treated sequentially with ARPI undergo any type of surgical resection/radioactive seed implantation? Did the previous surgical treatment bear any impact on which ARPI was used subsequently?
  5. Did the life expectancy improved with the sequential use of the ARPIs

Author Response

Referee 3 Comments and Suggestions for Authors

This manuscript reports a study on the use of ARPI in the treatment in CRPC in cancer patients. The study is potentially interesting but the enthusiasm for the topic is significantly limited by the manner the data were reported and discussed. Without a proper clarification and discussion of the points raised below no clear assessment of the significance of the results reported here can be properly made.

Comments:

  1. The authors reports that in their country the sequential use of ARPI is not recommended but yet, it appears that ARPI were used sequentially in all the years covered by the study. This begs the question: if the sequential use is not recommended, why these therapeutic agents were used sequentially? Insurgence of resistance? Recurrence of the cancer? Other reasons?

Response: Thank you for this good question! Unfortunately, while this study is based on register-based data, we are unable to assess the reasons for sequential ARPI use. Every physician is making independently their own clinical decisions in patient-care, based on their best of knowledge. Based on your good question we have added the following sentence in the Discussion section of the manuscript, please see lines 254–262.

This raises a critical question: why were these agents used sequentially despite the lack of proven efficacy? Possible explanations include avoiding of end-of-life discussions e.g. to palliative care after progression to one ARPI. Additional explanations could be influence from pharmaceutical marketing, management of adverse events, or perhaps a lack of awareness regarding current treatment guidelines. It is well known that today oncologist and urologists are inundated with information of new cancer medicine, clinical trial data and repeatedly updated treatment guidelines, which all together makes it difficult to stay up date, especially when national health care system covers the costs. “

2. Starting in 2020, there is a significant change in the ARPIs used (still sequentially, apparently) for the treatment of CRPC. Some explanation for the observed switch should be provided,

Response: Thank you! There might be many explanations for this observed change. First, some published guidelines may have influenced the sequential use and its prevalence. Secondly, it is possible that the uptake of the newest ARPIs, darolutamide and apalutamide have influenced use patterns of ARPI, as for instance the median duration of the treatment was longest for darolutamide (12.1 months). 

Based on your good comment we have added the following lines in the discussion section of the manuscript, please see lines 269–271

We also found that starting in 2020, there was a significant change in the ARPIs used and also sequential use diminished, which may be explained by the uptake of newest ARPIs, darolutamide and apalatumide and also the updated ESMO guidelines [18,19].

3. Where all the patients affected by CRPC placed on sequential administration of ARPI? If not, which patients were placed in this treatment category? And why? Did any of the patients considered here remain on a single ARPI treatment? And if so, what was the reason or rational for doing so?

Response: Thank you for this good question. Unfortunately, like described in the discussion section of the manuscript, in this register-based study we are unable to assess the rationale for any of the treatment decisions, such as which of the patients remain in single ARPI treatment and whose treatment was sequential or what was the stage of patient’s condition. However, your good question is an excellent suggestion for future research and based on it we have added following lines in the discussion section of the manuscript. Please see lines 317–318

“In addition, it would be worthwhile to assess how do patients with or without sequential ARPI use differ.”

4. Did any percentage of the patients treated sequentially with ARPI undergo any type of surgical resection/radioactive seed implantation? Did the previous surgical treatment bear any impact on which ARPI was used subsequently?

Response: We thank you for this important comment! However though, unfortunately this kind of assessment is impossible due to the information contained in our register. Based on your excellent comment we have included this as a limitation of the study, please see lines 309–310 in the discussion section of the manuscript as follows

“..ARPIs were used, what the previous prostate cancer treatments such as surgical resection/radioactive seed implantation and comorbidities..”

  1. Did the life expectancy improved with the sequential use of the ARPIs

Response: Thank you for this good question! As described earlier (please see Response to comment 8 of the Reviewer 2) we conducted a very crude Kaplan Maier analysis assessing the overall survival comparing patient using ARPIs sequentially and using them in monotherapy. We added the following paragraphs in the methods and results section, as follows:

Methods (lines 136–137): We conducted Kaplan-Meier analysis to compare the median overall survival (OS) among patients with single ARPI use and sequential ARPI use.

Results lines (190–192): During our study period of the patients using single ARPI 51% and of the patients with sequential ARPI use 82% had died. Based on Kaplan-Maier method the median OS among single ARPI users was 11.8 months and among sequential users 25.7 months.

 However, one limitation of our study is that we did not have in use register-based data containing information on the cause of mortality. As the majority of the patients included are older men, probably with many comorbidities, we did not assess detailed survival patterns, as the possible cause of death could be many other in addition to prostate cancer. We have also included the following sentence in the discussion section of the manuscript, please see lines 315–317:

We also missed the register-data concerning the cause of mortality in this study. Therefore, a future study is needed to assess how more detailed survival outcomes and life expectancy are related to different ARPI use patterns. “

Reviewer 4 Report

Comments and Suggestions for Authors
  • Clarity on clinical context

    • The introduction effectively reviews ARPI development but could be more concise and better structured around the rationale for studying sequential use specifically.

  • Methodological considerations

    • Sequential use definition is based on purchases with a 365-day gap. While practical, it may misclassify treatment interruptions or overlapping therapies. Clarify how overlapping prescriptions or treatment switches within short periods were handled.

    • Death data is included, but survival analyses are absent. Even descriptive survival outcomes (e.g., median overall survival among sequential vs. single ARPI users) would significantly enhance clinical interpretability.

    • Discussion

    • It may be useful to discuss how these findings should inform reimbursement policies and clinical decision-making.
    •  
  • I see a total of 30 references, but in the text only 27!?

Author Response

Referee 4 Comments and Suggestions for Authors

  • Clarity on clinical context
  1. The introduction effectively reviews ARPI development but could be more concise and better structured around the rationale for studying sequentialuse specifically.

Response: Thank you for this good comment! Based on it we have modified the last paragraph of the introduction section (lines 92–106) as follows

Annual healthcare costs per patient for metastatic prostate cancer are substantial and have increased over time, corresponding to the approval of new oral therapies used in treating metastatic prostate cancer [4–6]. Today, the number of PC patients is constantly increasing, and the patients live longer, leading to rapidly rising costs of cancer treatments and, consequently, challenges in access to treatment worldwide. [21,22]. Therefore, it is important to assess if there exists low value cancer care practices. Given the high prices of ARPIs, evidence on the real-world utilisation and costs of ARPIs is limited, especially regarding the sequencing of ARPIs in Scandinavia [23,19]. To our knowledge we lack knowledge on the population level on ARPI use patterns in real life and especially use patterns, which may have only little clinical benefit but high costs. To be better prepared to face these facts, we conducted a register-based study. The aim of this nationwide study was to assessed ARPI use patterns, especially sequential use and treatment costs.

 Methodological considerations

  1. Sequential use definition is based on purchases with a 365-day gap. While practical, it may misclassify treatment interruptions or overlapping therapies. Clarify how overlapping prescriptions or treatment switches within short periods were handled.

Response: Thank you for this excellent comment! This study was based on medicine dispense data, not data on prescriptions. In addition, we identified all new purchases of the ARPIs, suggesting that we have pretty well captured all initiated ARPI treatments and followed these purchases. However though, like described earlier, it is a limitation of this register-based study that we do not have recorded possible medicine switches (for instance due to adverse events). To assess this, we assessed the number of purchases of the first medicine among those persons, who used their ARPIs sequentially and the time to sequential use was less than a month. As a result, only 77 sequential users (3% of all sequential users) had purchased their first ARPI less than three times, suggesting that treatment switches were quite rare.

In addition to this, we have added a following limitation in the discussion section, please see lines 312–315

The utilized register does not contain any information on overlapping prescriptions or treatment switches within short periods, but treatment switches within a short period were e rare.  

3. Death data is included, but survival analyses are absent. Even descriptive survival outcomes (e.g., median overall survival among sequential vs. single ARPI users) would significantly enhance clinical interpretability.

Response: Thank you for this good question! As described earlier (please see Response to comment 8 of the Reviewer 2 and Comment 5 of the Reviewer 3) we conducted a very crude Kaplan Maier analysis assessing the overall survival comparing patient using ARPIs sequentially and using them in monotherapy. We added the following paragraphs in the methods and results section, as follows:

Methods (lines 136–137): We conducted Kaplan-Meier analysis to compare the median overall survival (OS) among patients with single ARPI use and sequential ARPI use.

Results lines (190–192): During our study period of the patients using single ARPI 51% and of the patients with sequential ARPI use 82% had died. Based on Kaplan-Maier method the median OS among single ARPI users was 11.8 months and among sequential users 25.7 months.

 However, one limitation of our study is that we did not have in use register-based data containing information on the cause of mortality. As the majority of the patients included are older men, probably with many comorbidities, we did not assess detailed survival patterns, as the possible cause of death could be many other in addition to prostate cancer. We have also included the following sentence in the discussion section of the manuscript, please see lines 315–317:

We also missed the register-data concerning the cause of mortality in this study. Therefore, a future study is needed to assess how more detailed survival outcomes and life expectancy are related to different ARPI use patterns. “

 Discussion

4. It may be useful to discuss how these findings should inform reimbursement policies and clinical decision-making.

Response: Thank you for this good comment! We have added the following paragraph in the discussion section of the manuscript, please see lines 301–307:

Our findings can inform reimbursement policies and clinical decision-making by offering an example why paying attention to rational use of medicines and especially to pharmacotherapy with low value care but high costs. Based on the statistics of SII ARPIs have been for many years among medicines with highest reimbursement costs in whole Finland due to their high prices and relatively large number users. Therefore, it is especially important to pay attention that high-cost medicines are used in a way that benefits patients most. 

  1. I see a total of 30 references, but in the text only 27!?

Response: We apologize the mess with the reference list. We have updated it and modified it throughout to better follow the instructions of MDPI, in addition we have double checked that it is correct.

Reviewer 5 Report

Comments and Suggestions for Authors

Title: Real-Life Use Patterns of Androgen Receptor Pathway Inhibitors (ARPIs): A Nationwide Register-Based Study in Finland during 2012 – 2023

The manuscript by Terhi et al. evaluated the androgen pathway inhibitor-based analysis in real-life use patterns with perspectives. Overall, the manuscript is interesting and requires revision as follows:

Comments:

  1. Line 46, the authors should provide some key information’s on the common cause of cancers, mortality, therapeutic approaches in correlation with prostate cancer contribution, i.e. doi: 3390/biomedicines11061611.
  2. Lines 85-87, the key objectives and significance of the present study should be clearly stated.
  3. The authors should provide in illustrations on the mechanism and significance of androgen pathway inhibitors.
  4. Discussion requires updates related with cost of other treatmemets approaches in prostate cancer (Provide literature summary Tables).
  5. Line 245-246, the major challenges and limitations of androgen pathway inhibitors-based treatment of prostate cancer should by summarized.

Author Response

Reviewer 5: Comments and Suggestions for Authors

  1. Line 46, the authors should provide some key information’s on the common cause of cancers, mortality, therapeutic approaches in correlation with prostate cancer contribution, i.e. doi: 3390/biomedicines11061611.

Response: Thank you for this suggestion! We have added the following sentence on lines 48–49:

Key causes of PC, similar to many other cancers, include genetic factors, exposure to different carsinogens and increasingly lifestyle related factors [3]

 In lines 57-62 therapeutic approaches are described

  1. Lines 85-87, the key objectives and significance of the present study should be clearly stated.

Response: Thank you! Based on your good comment, we have modified the last paragraph in the introduction as follows, please see lines 92–106 in the current version of the manuscript.

Annual healthcare costs per patient for metastatic prostate cancer are substantial and have increased over time, corresponding to the approval of new oral therapies used in treating metastatic prostate cancer [4,5]. Today, the number of PC patients is constantly increasing, and the patients live longer, leading to rapidly rising costs of cancer treatments and, consequently, challenges in access to treatment worldwide. [20,21]. Given the high prices of ARPIs, evidence on the real-world utilisation and costs of ARPIs is limited, especially regarding the sequencing of ARPIs in Scandinavia [19]. To our knowledge we lack knowledge on the population level on ARPI use patterns and especially those, which may have only little clinical benefit but high costs. To be better prepared to face these facts, we conducted a register-based study. The aim of this nationwide study was to assessed ARPI use patterns, especially sequential use and treatment costs, in this nationwide study in 2012–2023. 

  1. The authors should provide in illustrations on the mechanism and significance of androgen pathway inhibitors.

Response: Thank you! As there is an excellent illustrations on the mechanism and significance of androgen pathway inhibitors in reference number 10 we have referred to it in the main text as follows. Please see lines 70–71 in the introduction

“A detailed illustration on the mechanism and significance of ARPIs is presented in [10], for instance”

  1. Discussion requires updates related with cost of other treatmemets approaches in prostate cancer (Provide literature summary Tables).

Response: Thank you! Unfortunately, as we lack detailed information from Finland concerning the costs of other treatments, such as resection, radiotherapy, etc. and therefore we are unable to compare the costs of ARPIs to them. While the health care systems widely differ between countries, we are unable to compare Finnish situation to foreign data. However, to better respond to your good comment, we utilized AI and made a brief literature summary, as a supplement of this response letter, covering this matter, based on different publications. To note, we have not utilized AI while conducting or reporting this study, but for this specific purpose to answer your question we used AI-assisted literature search and utilized Elicit® for this purpose.

  1. Line 245-246, the major challenges and limitations of androgen pathway inhibitors-based treatment of prostate cancer should by summarized.

Response: Thank you for this comment! We have modified the following sentence and added some most common limitations of ARPI-treatment in currently lines 296–300 such as follows:

Preventing futile treatments in palliative patients enables better symptom manage-ment without the ARPI side effects, such as anemia, hot flushes, hypertension, fatigue, musculoscletal pain for instance [10,32] reducing the financial impact of cancer care. In addition to adverse events, also drug interactions, drug resistance and impact on functional capacity should also be taken into account as major limitations of ARPI treatment.

Round 2

Reviewer 2 Report

Comments and Suggestions for Authors

Authors have satisfactorily responded to all my concerns and accordingly improved quality of this manuscript.

Author Response

Comments and Suggestions for Authors

Authors have satisfactorily responded to all my concerns and accordingly improved quality of this manuscript.

Reply: Thank you for your valuable comments, which truly improved the manuscript!

Reviewer 3 Report

Comments and Suggestions for Authors

This is a resubmission of a manuscript reporting a study on the use of ARPI in the treatment in CRPC in cancer patients. The study is interesting and the major criticisms raised at the time of the initial submission have been addressed satisfactorily in this revision. 

Author Response

omments and Suggestions for Authors

This is a resubmission of a manuscript reporting a study on the use of ARPI in the treatment in CRPC in cancer patients. The study is interesting and the major criticisms raised at the time of the initial submission have been addressed satisfactorily in this revision. 

Reply: Thank you for your valuable comments and suggestions, which truly improved this manuscript! 

Reviewer 4 Report

Comments and Suggestions for Authors

authors have addressed my major concerns

Author Response

Comments and Suggestions for Authors :authors have addressed my major concerns   Reply: Thank You so much for your very valuable comments and suggestions! You helped us a lot in improving the manuscript! 

Reviewer 5 Report

Comments and Suggestions for Authors

Accept

Author Response

Comments and Suggestions for Authors: Accept     Reply: Thank you for your good comments and suggestion for this manuscript and your help in improving it!